# Lignocellulosic Biomass as Sorbent for Fluoride Removal in Drinking Water

**DOI:** 10.3390/polym14235219

**Published:** 2022-11-30

**Authors:** Adriana Robledo-Peralta, Luis A. Torres-Castañón, René I. Rodríguez-Beltrán, Liliana Reynoso-Cuevas

**Affiliations:** 1Department of Sustainable Engineering, Advanced Materials Research Center (CIMAV-Durango), CIMAV 110 Street, Ejido Arroyo Seco, Durango 34147, Mexico; 2CONACYT-Centro de Investigación Científica y de Educación Superior de Ensenada, Unidad Foránea Monterrey, Alianza Centro 504, PIIT, Apodaca 66629, Mexico; 3CONACYT, Advanced Materials Research Center (CIMAV-Durango), CIMAV 110 Street, Ejido Arroyo Seco, Durango 34147, Mexico

**Keywords:** adsorption, innovative materials, water treatment, biocomposite, fluoride

## Abstract

Water supply to millions of people worldwide is of alarmingly poor quality. Supply sources are depleting, whereas demand is increasing. Health problems associated with water consumption exceeding 1.5 mg/L of fluoride are a severe concern for the World Health Organization (WHO). Therefore, it is urgent to research and develop new technologies and innovative materials to achieve partial fluoride reduction in water intended for human consumption. The new alternative technologies must be environmentally friendly and be able to remove fluoride at the lowest possible costs. So, the use of waste from lignocellulosic biomasses provides a promising alternative to commercially inorganic-based adsorbents—published studies present bioadsorbent materials competing with conventional inorganic-based adsorbents satisfactorily. However, it is still necessary to improve the modification methods to enhance the adsorption capacity and selectivity, as well as the reuse cycles of these bioadsorbents.

## 1. Introduction

The element fluorine (F) has been described over practically the entire Earth’s crust. It is the fundamental reason for its presence at various concentrations in the waters around the world [1,2]. In aqueous media, fluoride is present as fluoride anion (F^−^) [1]. In humans, the primary source of fluoride intake is drinking water [2]. The World Health Organization (WHO) establishes a Maximum Permissible Limit (MPL) for fluoride in drinking water of 1.5 mg/L. It estimates that water intake in concentrations above this limit represents a health threat to millions worldwide [3,4]. In addition, dental and skeletal fluorosis, neuronal damage, and fertility issues are recognized as health problems associated with fluoride ingestion [1,2,5].

The WHO also recommends fluoride concentrations ranging from 0.5 to 1 mg/L in drinking water as a preventive measure to avoid the incidence of dental caries. However, to prevent damage to the population’s health, it is necessary to remove this element from the water up to harmless concentrations [2].

For developing countries, the WHO reports that drinking water defluoridation technologies are generally applied in batch or batch flow systems, at household or small community scale, using conventional technologies such as precipitation, activated alumina, bone charcoal, clays, and other naturally occurring media. In contrast, for industrialized countries, continuous systems and advanced treatment technologies such as reverse osmosis and electrodialysis are used [5].

Adsorption has been extensively studied and recognized as an efficient and cost-effective alternative for drinking water treatment [3,6,7]. The solid-particulate material that captures the compounds of interest is an adsorbent [8,9]. To date, the available diversity of adsorbents is vast. Among them, materials of renewable sources, such as plant tissues rich in lignocellulosic composition, have emerged with great potential [10,11]. Non-carbonized lignocellulosic materials are a lower-cost option [6,9], and cellulose is a biopolymer rich in hydroxyl [12,13]. Moreover, hydroxyl groups can adsorb fluoride and other anions through hydrogen-bonding interactions [14]. Furthermore, cellulose can be used as a matrix for metal cation impregnation due to electrostatic interactions between the hydroxyl group and the cations, thus increasing its adsorption capacity [9,14].

Therefore, several investigations document non-carbonized plant materials with competitive removal capacities. These materials have been tested to remove pollutants such as As, Co, Cu, Hg, Cd, Zn, Ni, Cr, Pb, F, NO^−3^, SO_4_^−2^, CN^−^, PO_4_^−3^, dyes, and even pharmaceuticals, all of them in an aqueous media [15,16,17,18,19,20,21,22,23,24,25,26,27,28,29,30].

## 2. Technologies for Fluoride Removal

Different technologies have been studied to remove fluoride excess in drinking water. However, selecting a treatment method will depend on technical and economic feasibility and the water to be treated. Table 1 shows a general overview of the technologies used for pollutant removal and describes the method and the advantages and disadvantages of each one.

Adsorption is the leading technology for the pollutant removal of liquid effluents. It has been widely studied and used due to its simple design, low costs, and simple operation [31]. Among the adsorbent materials, using modified lignocellulosic biomasses represents an interesting option for fluoride removal, as the efficiencies achieved compete with more expensive chemical processes.

## 3. Fluoride Removal by Biosorption

A biomass-based adsorbent is called a biosorbent. Thus, biosorption is a physicochemical phenomenon that refers to the ability of the biosorbent to capture ions present in an aqueous solution. The biosorbent can be used in a pristine form or modified as a biocomposite [13,35]. Additionally, they can be used as dry or calcined matter (activated carbons). Bioadsorbent materials can be classified into chitin and chitosan, microorganisms (bacteria, algae, and fungi), lignocellulosic plant matter (agroindustrial and food waste), and animal material (bone).

Lignocellulosic biomass is the dry matter of plants (biomass) and is the most abundant raw material on Earth. In many countries, lignocellulosic biomass, such as fruit peels, agricultural residues, and garden waste, is part of solid municipal waste [36]. Depending on the plant type, vegetal tissues are mainly constituted of lignin (10–25%), hemicellulose (20–35%), and cellulose (35–50%) of the dry biomass [13]. The lignocellulosic biomass provides a significant content of functional groups and available sites for fluoride capture [6,36]. Specifically, cellulose has excellent adsorption capacities due to the abundance of the hydroxyl group [37].

Huang et al. (2022) [36] stated that activated carbons and adsorbents modified with metal ions have the best adsorption efficiencies, and a diversity of mechanisms to adsorpt.

In the last ten years, the literature has reported pollutant ion removal from water by dry and modified lignocellulosic biomasses such as Zr-loaded orange waste gel [38], apple pomace and peel [19,22], wheat straw, and sawdust [39], pine sawdust [39], lemon leaves [40], several fruit peels such as banana, grape, peach, and cassava [24,41,42], tea leaves [43], jute fibers [44], guava seeds [45], sisal fiber [46], garlic husks [47], and tamarind seeds [48], among others.

In most studied biomasses, cellulose is the principal active agent, a natural polysaccharide with hydrogen bonds and good stability [12,20]. Therefore, the biopolymer has been studied as a matrix for elements impregnation through electrostatic interaction with the cellulose hydroxyl groups. In addition, these groups could also be used for anion adsorption by hydrogen-bonding interactions on the biopolymer surface [13,14,20].

### 3.1. Fluoride Removal from Drinking Water by Lignocellulosic Biomasses

Many materials of plant sources have been studied for fluoride and other pollutant ion removal from water. This research focused on lignocellulosic biomasses that have not been thermally treated and have not been carbonized. Table 2 shows an overview of bioadsorbents used to remove fluoride ions from drinking water, including pure and chemically modified biomasses.

### 3.2. Methods for Bioadsorbent Preparation

Lignocellulosic biomasses need to be processed before being applied as bioadsorbents. The procedures can be simple, such as drying, grinding, and sieving (physical treatment) to obtain fine powders. However, they can also be more sophisticated and require reagents and laboratory equipment (chemical treatment). Using one or more physicochemical treatments is common to improve the fluorine capture capacity and selectivity. However, not only do they help to improve the capacity, but they also sometimes serve to remove interfering or unwanted compounds [32]. These can be water-soluble molecules, pigments, essential oils, or others.

Generally, in the final stage of chemical treatment, plant tissues undergo a transformation, which makes it possible to subject them to impregnation or doping with metal cations linked to the pollutant of interest. Figure 1 describes how lignocellulosic materials are processed for their application as bioadsorbents. However, not all of them necessarily occur, except drying and particle size reduction.

#### 3.2.1. Physical Treatments

Physical processes include tap water washing and further washing with distilled, double-distilled, and/or deionized water. This is intended to remove dust, dirt, and other impurities from the biomasses, avoiding interference with the remaining processing [14,38,40,42,45,47,49,50,58]. Usually, it is the initial stage of the preparation process. However, it does not occur in all cases [39,43,44].

Table 3 shows the drying temperatures and particle sizes reported in biomass processes used as bioadsorbents. Mainly, drying temperatures are low; this is due to the need to remove only the moisture (water), so exceeding 100 °C at ambient pressure is unnecessary. As the grinding method, using a mortar was the most common; however, reducing the particle size in a mill may be more practical. Particle sizes range between 75 and 1500 μm, with larger particles being easier to handle and separate in the adsorption process; however, these have less surface area, and therefore fewer active sites are assumed.

#### 3.2.2. Chemical Treatments

After physical treatments, some authors subject plant tissues to chemical treatments. The purpose of such treatments can be the elimination of pigments, destruction of compounds, and bond formation or destruction, among others. In other cases, these treatments turn the biomass into an element-bearing material (biocomposite). Biocomposite formation is intended to increase the adsorption capacity of biomass.

Paudyal et al. (2012a) subjected orange waste to saponification with calcium hydroxide, and then the prepared white material was transformed into a cation-exchange gel containing calcium. Subsequently, this gel was placed in contact with zirconium oxychloride octahydrate, thus completing the preparation and loading Zr into the biomass base [38].

Additionally, Paudyal et al. (2012b), in similar research, used saponified orange peels to impregnate them with different cations. During the loading reaction, the presence of Ca^+2^ ions in the saponified product undergoes a cation exchange reaction with the charged metal ions [49]. This promotes the loading of cations in the organic structure by forming stable chelates.

Subsequently, Paudyal et al. (2013) studied orange juice residues. Their research described the importance of water-soluble organic acid removal, such as citric acid, as these substances remove the previously charged metal ions [62].

Yadav et al. (2013) immobilized color and water-soluble substances by treating sawdust and wheat straw with formaldehyde (1%) in a 1:5 ratio (material: formaldehyde, *w*/*v*) at 50 ◦C up to 4 h [39].

Cai et al. (2015) removed tea-colored residues and soluble components by acid treatment with sulphuric acid (1 M). After the acid treatment, the acidified biomass was loaded with Fe and Al 0.4 salt solutions and adjusting the pH with sodium hydroxide (2 M) [43].

Jha et al. (2015) studied fluoride adsorption by Zr-loaded carboxylated orange peel. This study reported that orange waste contains cellulose, hemicelluloses, lignin, and pectin; the latter is estimated to contain 10% of the total orange waste. Pectin is a polysaccharide consisting mainly of hydroxyl and methyl ester groups. Because pectin’s ester and hydroxyl groups do not bind significantly with metals, the authors modified the orange peel by ester group saponification with sodium hydroxide, followed by hydroxyl group carboxylation with chloroacetic acid, thus increasing the number of carboxylate ligands. The Zr-binding capacity of the modified orange peel increased the biomass that can adsorb fluoride from an aqueous solution via a ligand-exchange mechanism [51].

Manna et al. (2015) performed an alkali-steam treatment to treat their lignocellulosic biomasses. Their publication stated that the modified biomass adsorption capacity was better than before the treatment [50]. Later on, Manna et al. (2018) found that alkali-steam treatments remove part of the amorphous cellulose, hemicellulose, and lignin, increasing the micro- and macropores on the bioadsorbent surface and thus, allowing access to more functional groups within the biomass, and increasing fluoride uptake [32].

Naga Babu et al. (2018) performed a combining of the biomass with hydrochloric acid to activate the sorption sites and leaving it in a water bath for 2 h [58].

Mwakabona et al. (2019) impregnated the biomass with Fe(III) by immersing it in Fe(III) chloride. They attributed the Fe charge on the particle surface to the coordination bonds provided by the electrostatic forces between the biomass surface and the Fe(III) complex in the solution. Subsequently, a portion of the impregnated material was soaked in a sodium hydroxide solution (post-alkalinization) to accomplish the Fe(III) co-precipitation forming Fe(III) hydroxide-coated surfaces. The adsorption experiments showed differences between the two materials. The non-alkalinized material was effective over a wide pH range, while the post-alkalinized material was more chemically stable [46].

Carboxylation is another common technique used to enhance the biomass defluorination capacity, which theoretically increases the fluoride ion removal rate and the pH of the reaction medium [63]. For instance, the number of carboxylate ligands has an increment because of the hydroxyl group carboxylation with chloroacetic acid. This modification favors the capacity of the biomass to bind metal cations. So, it can then adsorb fluorides from the aqueous solution by ligand exchange [51].

A literature review allows us to identify that the main cations used in the biomass impregnation that adsorb fluorides are La, Ca, Fe, Al, Ce, and Zr [14,37,38,42,45,46,47,49,51,54,56,62,64,65]. Each of them has its advantages and disadvantages. Namely, La and Zr have been described as stable and environmentally safe (non-toxic), selective, and as having high affinities for fluoride [54,66]. Meanwhile, Fe and Ca show good removal capacities but generally do not outperform La and Zr (Table 1). Finally, Al has a high affinity for fluoride. However, the working pH and fluoride dissolution in the treated water is debatable in their application [20,31,43].

Table 3 shows some chemical processing methods for biomasses used as fluoride bioadsorbents. The acidification or alkalinization of the lignocellulosic material allows the undesirable compounds’ elution from the material; this could prevent the modification of color, taste, or other characteristics in the treated water. It also enables cation doping to bind to stable compounds and not to lodge in water-soluble compounds, which would eventually be lost in the water treatment. In this case, acid treatments are expected to activate the adsorption sites. Moreover, alkaline treatments transform the raw material, e.g., the cellulose crystallinity changes and the roughness increases, among others, thus favoring the interaction with the adsorbate. Furthermore, intermediate treatments help prepare the biomasses when cations are accommodated in their structure. Finally, cation loading of biomolecules increases the capacity and selectivity of bioadsorbents, but increases costs, depending on the cation selected. 

Other alternative treatments include cellulose extraction, which increases adsorption capacity, and material pyrolysis, which increases porosity and surface area. However, this involves higher costs due to energy and reagent consumption and eventual waste generation [29,35,36,53,67].

## 4. Biosorbents Characterization

The characterization of base materials (biomass and reagents), intermediate materials, and final materials (biocomposites and biosorbents used) is a critical stage in understanding the nature of bioadsorbents, as well as the governing mechanisms for capturing the pollutant of interest [9]. Moreover, it is also essential to determine bioadsorbent stability and the quality of the treated water. Below is a brief description of frequently used techniques for lignocellulosic-based adsorbent materials characterization (Figure 2).

### 4.1. ICP (Inductively Coupled Plasma)

ICP is a practical, versatile and effective technique for the quantification of a large number of elements [68].

Paudyal (2013) determined the Zr, Al, and Ce concentration loaded in orange peel by ICP/AES. Additionally, he used this method to quantify the amount of residual Zr in treated water. The results showed that the Zr leaching is insignificant at pHs between 4 and 10, even when the initial fluoride concentration is up to 120 mg/L [62].

Jha (2015) quantified the concentration of Zr by ICP-OES in his orange peel bioadsorbent. He found that the Zr content in his bioadsorbents was 0.38 g Zr/g [51]; this value is higher than that reported for Zr impregnated orange waste by Paudyal (2012), which was 0.147 g Zr/g [38].

Likewise, Cai (2015) evaluated the concentration of dissolved Fe and Al in treated water by ICP-MS. The reaction medium pH significantly influenced the material’s stability. Residual amounts of Fe and Al were observed in the treated water. However, Fe and Al quantification showed that the concentration of Fe and Al remained below the permissible limits established by the WHO [5].

### 4.2. TGA/DSC (Thermogravimetric Analysis—Differential Scanning Calorimetry)

A thermal analysis (TA) allows relating temperature to the specific physical properties of a material.

Jha (2015) determined the amount of organic and inorganic matter in his orange peel bioadsorbent. The weight loss observed between 25 and 185 °C can be attributed to the evaporation of chemically and physically adsorbed water molecules and other volatile chemicals. Moreover, pectin, hemicellulose, and cellulose degradation could be related to weight loss at 185 to 300 °C. Finally, the weight loss in the 300 to 550 °C range is due to lignin degradation. The total weight loss in the bioadsorbent was 97.21% within the temperature range studied, leaving 2.79% of metallic residue due to the Zr impregnation [51].

### 4.3. SEM-EDS (Scanning Electron Microscope with Energy Dispersive Spectroscopy)

The microstructure and morphological characteristics of the adsorbent particles can be determined by SEM analysis [9,48].

Jha (2015) analyzed the surface morphology of his material (orange bioadsorbent) using a SEM with the energy dispersive spectroscopy (EDS) technique. The fluoride adsorption on the bioadsorbent can be associated with the presence of white spots distributed throughout the material, which is confirmed by EDS spectra [51].

Mukherjee et al. (2016) demonstrated by SEM analysis that the dry biomass of taro (*Colocasia esculenta*) has low porosity, while the activated carbon made from the same biomass has a highly developed porous structure. In both samples, the existence of fluoride was confirmed by EDAX analysis The adherence of fluoride was lowest in the dry biomass [53].

Zhang et al. (2019) used SEM-EDS analysis to observe Zr-loaded grape debris’s surface morphology and chemical composition before and after processing and contact with fluoride [42]. Furthermore, it was found that the unprocessed powder particles consist of C, Cl, and O with a porous surface suitable for Zr loading. After loading, the EDS spectra showed significant amounts of Zr forming active sites on the bioadsorbent surface. Additionally, by EDS, successful fluoride capture was determined.

Srinivasulu et al. (2021 a) studied tamarind seed husk bioadsorbent morphology by SEM-EDS analysis. They found that the biomass had an opaque appearance, with a non-adhesive nature. The particles had sharp edges and a characteristic irregular structure. In addition, it was determined that the microporous particles possessed voids and provided sites for fluoride capture. After biosorption, the adhesion of fluoride ion particles appeared in EDX analysis, which also showed the presence of O, C, Ca, and Mg in the untreated materials. Finally, the elemental oxygen concentration was reduced after biosorption and fluoride was introduced, suggesting that the fluoride anion can replace the ion-containing oxygen atoms (OH) [48].

Yao et al. (2021) performed SEM analysis on samples of base, intermediate, and final materials before and after fluoride contact. The results allowed them to describe the rough texture and the irregular pore structure of the bioadsorbent material, an ideal characteristic for fluoride adsorption since it provides sufficient surface area and active sites to achieve a high adsorption capacity. Additionally, the sample that adsorbed fluoride showed no significant change compared to the one that did not. This denotes its stability and potential application in water treatment. The mapping of the elements C, O, and Ce on the bioadsorbent surface, before and after fluoride capture, showed homogeneous distribution over the material surface, suggesting a successful preparation. Fluoride was also detected in the EDS analysis [37].

### 4.4. BET (Brunauer, Emmett, and Teller)

Knowing the surface characteristics is valuable for describing its potential use. One of the fundamental properties is the surface area available for molecule adsorption.

Jha (2015) determined the surface area of his bioadsorbent by BET using the Quantachrome automated gas sorption system. The bioadsorbent specific surface area, micropore volume, and mean micropore diameter were 20.56 m^2^/g, 0.0001 cm^3^/g, and 2.62 nm, respectively [51].

The surface area of a tamarind seed-husk was established by Srinivasulu (2021a) using BET at 286.94 m^2^/g. In contrast, the bioadsorbent area was 93.38 m^2^/g. The authors attributed the surface area reduction to the fluoride sorption on the biosorbent surface [48].

Mukherjee (2016) measured the taro stem (*Colocasia esculenta*) bioadsorbent surface area and pore volume, and these values were compared with carbonized and steam-activated taro stem samples. The results showed that both pieces are microporous; however, all characteristics were superior in the activated carbon. The results proved why the dry biomass had lower fluoride removal compared with that of the activated carbon from the same biomass. [53].

Srinivasulu et al. (2021b) determined the surface area of the *Senna auriculata L* flowers petals by BET as 239.94 m^2^/g; the area of the petals processed as bioadsorbent was 286.94 m^2^/g, and that of the fluoride adsorbed sample was 97.68 m^2^/g. These changes are attributed, as in the case of the tamarind seed husk, to the fluoride capture on the bioadsorbent [61].

### 4.5. FTIR (Fourier Transform Infrared Transmission Spectroscopy)

The IR spectrum of functional groups is considered unique and characteristic of the molecule.

A given absorption band assigned to a functional group proportionally increases with the number of times that functional group occurs within the molecule. FTIR is often used for qualitative biopolymer characterization [69,70,71].

Jha (2015) obtained the FTIR spectra of Zr-impregnated orange peel before and after contact with fluoride. In the fluoride-captured bioadsorbent spectra, a broad peak at 3569 cm^−1^ shows the presence of -OH groups on the surface. A peak at 496 cm^−1^ in the same sample indicates the presence of Zr-O bonding. Furthermore, the peak at 3569 cm^−1^ in the case of the unused bioadsorbent was shifted to 3615 cm^−1^ in the sample corresponding to the used bioadsorbent. This shows the participation of the hydroxyl anion in the ligand exchange with the fluoride [51].

Peng et al. (2017) investigated the fluoride adsorption mechanism through the spectra of Al-impregnated tea waste adsorbent. They observed that the band at 3377 cm^−1^ shifted to 3424 cm^−1^ and broadened, due to Al impregnation, leaving evidence of the metal ion interaction with the OH groups. Furthermore, upon adsorption of the fluoride, the 880 cm^−1^ peak appeared, which was attributed to Al–F stretching vibrations, suggesting adsorption by ion exchange with the OH bonded to Al [55].

Zhang (2019) presented the FTIR spectra of Zr-modified grape pomace. The biomass showed peaks related to stretching vibrations of -OH, -CH_2_, COO, and -C=O groups at wavelengths 3423, 2927, 1739, and 1631 cm^−1^, respectively. However, the latter shifted to 1625 cm^−1^ after Zr loading. This could be caused by the exchange of Zr ions with the hydrogen present in the base biomass and shows the fundamental role of carbonyl functional groups in the Zr impregnation of the biomass. Subsequently, Zr forms hydrated compounds with high fluoride affinity [42].

Srinivasulu (2021a) described the presence of different functional groups in biomass, which are imputed to the nature of it. For instance, at wavelength 3417.88 cm^−1^, the hydroxyl functional group associated with organic acids, alcohols, and phenols was identified. Furthermore, the stretching of the carbonyl group was identified at 1620.36 cm^−1^. These are more pronounced in the oxidized carbon materials than in the original ones. In addition, broad and long peaks were found on the surface of the bioadsorbent, possibly due to the chemical activation process. That is, the hydrogen ion released by the sulphuric acid could lead to the appearance of the hydroxyl functional group on the surface of the bioadsorbent [48].

Yao (2021) confirmed the surface composition of the material as well as the mechanism by which the fluoride is attached by FTIR analysis. The bioadsorbent spectra showed peaks attributable to the vibrations of the C–H, C–O, C–O–C, and Ce–O–Ce bonds. The fluoride retained material spectrum also showed the presence of peaks below 600 cm^−1^, consistent with Ce–F bonds. The 3325 cm^−1^ peak, attributed to the stretching of hydroxyl groups, indicated the abundant presence of hydroxyl groups. Upon fluoride adsorption, the band was reduced, as evidence of the critical role of hydroxyl groups. Based on the above, the authors suggest that the mechanism of fluoride ion adsorption is the substitution of species between the fluoride and hydroxyl groups [37].

### 4.6. XRD (X-ray Diffraction)

XRD patterns allow for compound identification. For instance, the literature reports biocomposite and biomass characterization by X-ray diffraction, determining semi-crystalline characteristics due to molecules such as cellulose or lignin.

Jha (2015) evaluated Zr-modified orange peel. The XRD analysis revealed the presence of Zr(OH)_4_ and polymerized species. It was indicated that the charged Zr(OH)_4_ is transformed into an amorphous phase composed of polymerized species by water incorporation [51].

Mukherjee (2016), through dried and charred taro (*Colocasia esculenta*) stem sample diffraction patterns, showed few sharp peaks before and after adsorption, which shows that the samples are crystalline. Furthermore, after the fluoride ions adsorption, the peaks intensity decreased, indicating the adsorption of fluoride ions on the surfaces of the crystalline structure by physisorption [53].

Nagaraj et al. (2017) extracted cellulose from sugarcane bagasse by ultrasound, which they modified with La. The cellulose and bioadsorbent were analyzed by X-ray diffraction. The analysis showed that the ultrasound treatment weakened the hydrogen bonds between the cellulose layers, thus forming individual layers, which led to increased interaction with the La particles. As a result, La could fill the intermediate layer of the biopolymer matrix during ultrasonication. The 20°–29° peaks indicate La impregnation in the cellulose, and the peaks at 25.86° and 49.56° are due to fluoride adsorption on the bioadsorbent [54].

Yao (2021) demonstrated the CeO_2_ loading on the cellulose membrane used for fluoride adsorption under XRD analysis. He confirmed the cellulose extraction success in the wood residues. The analysis allowed him to identify not only the compound but also its purity [37].

### 4.7. XPS (X-ray Photoelectron Spectroscopy)

XPS analysis is not only helpful in determining the composition of composite materials but has also been shown to be helpful in determining the mechanism by which fluoride is adsorbed on such materials.

Manna (2015) demonstrated, by XPS spectra, the binding of fluoride and jute bioadsorbent. In the C1s spectra, the C2 (carbon C–OH), C3 (carbons O–C–O or C=O), and C4 (carbon O–C=O) peaks shifted to higher binding energy ranges, and the atomic percentages of these carbons decreased after exposure to the fluoride solution. This was attributed to hydrogen bonding between fluoride ions and OH groups or OH group substitution by fluoride ions and electrostatic interaction between fluoride ions and protonated carbonyl groups [44].

Peng (2017) determined the fluoride ion-capture mechanism by observing a significant increase in F peak area (0.54–1.22%) after adsorption and associated the change to the surface reaction between the Al–Tea biosorbent and the fluoride. Additionally, changes in Al(2p) peak shapes and binding energy were observed after adsorption, implying that Al participated in the adsorption of fluoride [55].

Yao (2021) confirmed the fluoride adsorption mechanism by XPS peak deconvolution for the elements C, O, and Ce present in the bioadsorbent samples before and after contact with fluoride. In the results, it was identified that Ce coexists in the Ce^+3^ and Ce^+4^ oxidation states. The Ce peak was not detected after fluoride adsorption due to fluoride capture. The C1s showed a reduction in the peak area of the C–O and C–C, which is attributable to the presence of fluoride on the surface. The peak shift and significant decrease of the Ce–OH area in the O1s spectrum and the C and Ce spectra give evidence of fluoride capture success in the study material. So, the peak deconvolution, which showed shifts in the peaks and changes in intensities and area, leaves the certainty of fluoride capture by ion exchange between the hydroxyl group and the fluoride [37].

The quantification of the elements supported in biomasses and the validation of their concentration in treated water (material stability) have not been commonly explored, despite their importance. Quantifying elements in water by ICP is a fast and accurate option; this analytical method is also very useful for quantifying elements in a complex matrix, such as bioadsorbents. EDX, EDS, and EDAX analysis can also determine so-transported elements in biomasses.

BET is another uncommon method in the adsorbent bio-material characterization. However, describing both surface area and pore size is of utmost importance to explain firstly the viability of the material for capturing the pollutant of interest and the mechanism by which adsorption takes place. Finally, thermogravimetric analyses are another analytical method rarely used to characterize bioadsorbents. Although they do not describe the adsorption capacity or type of adsorption performed by biosorbent materials, these techniques describe the composition and the changes that the base materials undergo in their structure and surface until they become bioadsorbents.

The most essential and commonly used analysis techniques to describe bioadsorbent materials are FTIR and SEM. Many authors describe both the composition in functional groups and the presence of cations bonded to the matrix of an organic nature, and some works even describe bond formation with fluoride. By SEM analysis, the authors generally report the texture of the materials, as well as the shape and size. This analysis and BET can be an enriching complement to relate the morphology of adsorbent particles and thus explain the interaction between fluoride anions dissolved in water and the bioadsorbent.

Finally, the adsorption mechanism could be explained by XRD and XPS analysis. In them, it is possible to establish the material composition in a detailed matter, thus allowing observing changes in the oxidation states and the formation of bonds and compounds during the modification of the bio-masses until the fluoride is captured.

The analysis selection depends on the base material nature, the information needed, and the available resources. However, it is necessary to implement the same analysis for base, intermediate, and final materials. This process will allow the identification and description of all the transformations undergone at each stage.

## 5. Biosorbents Ionic Affinity

The anion and cation coexistence during fluoride adsorption can reduce the ability of the material to capture fluoride due to precipitation, complex formation, or competition for sorption sites [56]. Therefore, it is necessary to determine the selectivity, described as the adsorbate–adsorbent affinity and interferences (species competing with the contaminant of interest for active sites) of the bioadsorbents. These factors are crucial to verifying the viability of bioadsorbents for use in natural, waste, and drinking water.

Water intended for human consumption has different anion and cation concentrations, depending on the subsoil composition with which it interacts [1,5]. However, most ions determine the type of water in question [5,72]. Therefore, the main constituents of groundwater include those described in Table 4.

Specifically, the fluoride adsorption in the presence of anions at different concentrations must be studied to evaluate the bioadsorbent selectivity and the competition between ionic species and fluoride for active sites [51,56]. The literature reports that carbonates (CO_3_^−2^) and phosphates (PO_4_^−3^) mainly affect fluoride adsorption. However, other ions may also interfere with fluoride removal. Table 2 shows the ionic interferences reported for fluoride adsorption.

Paudyal (2012) developed an adsorbent with orange peels. In his study of coexisting ions, he found no significant reduction in fluoride removal in the presence of ions Cl^−1^, NO_3_^−1^, CO3^−2^, and SO_4_^−2^ under a pH range of 2–4 [38].

Jha (2015) reported that for his orange adsorbent, HCO_3_^−1^ and PO4^−3^ concentrations above 200 mg/L decreased the fluoride removal percentage below 90%, and at 600 mg/L, fluoride adsorption dropped to less than 70%.

Kazi et al. (2018) determined that SO_4_^−2^ anions severely negatively affect fluoride bioadsorption by melon peels. Likewise, the effect was associated with a higher negative charge compared with that of Cl^−^ and Br^−^, whose interference was slight [56].

Zhang (2019) showed that SO_4_^−2^, NO_3_^−^, and Cl^−^ ions did not interfere with fluoride biosorption by Zr-modified grape bagasse. However, HPO_4_^−2^ and CO_3_^−2^ ions show considerable interference in fluoride removal [42].

The nature of the base material such as the functional group richness and the presence of fluoride-related cations are crucial factors in the biosorbent material selectivity. The anion interference degree on fluoride adsorption also depends on the operating conditions, e.g., pH. This is extremely important when testing biosorbents on real materials. The feasibility of a material can be undermined if the water composition and operating conditions that favor the capture of fluoride anions are not considered. In addition, treatment costs may increase as higher doses of bioadsorbent may be required.

## 6. Materials Description and Reuse Cycles

Material desorption (elution) is the reverse process of adsorption. The aim is to release or separate, in this particular case, the fluoride from the bioadsorbents [38,49,62]. The procedure is intended to reuse or dispose of the adsorbent material safely [37,48]. Although reuse is intended to increase the material lifetime, it must also be considered that, in practice, it contributes to cost reduction. Therefore, it is essential to study the desorption medium and desorption conditions.

Different materials are reported in the literature to have undergone desorption, and in general, this procedure was carried out in a basic medium, using NaOH. Table 2 summarizes bioadsorbents with desorption cycles in batch and continuous systems, highlighting materials that achieved 8 to 10 desorption cycles. Finally, NaOH is the leading reagent used for bioadsorbent desorption.

## 7. Mechanism

Adsorption consists of capturing or retaining an adsorbate present in the aqueous or gas phase on a porous solid material known as an adsorbent. During adsorption, physical or chemical bonds may occur between the adsorbent particles and the adsorbate. The strength of interaction between the two will define whether physisorption or chemisorption occurs [73].

Al-Ghouti et al. (2020) stated that in physisorption, multilayer formation occurs and is attributed to weak electrostatic interactions such as London forces, dipole–dipole forces, and Van der Waals interactions. Conversely, monolayer formation takes place in chemisorption due to the forming of bonds between adsorbate and adsorbent by sharing or transferring electrons. Chemisorption interactions are two orders of magnitude than physisorption [73].

Different adsorbent characterization techniques, before and after adsorption, and equilibrium data modeling (isotherm and kinetic models) has been studied at a constant temperature to describe the interaction mechanisms between adsorbate and adsorbent [73,74].

Numerous isotherm models have been developed and studied, and each of them involves different parameters and implies various considerations/conditions. However, the Langmuir and Freundlich models are the most widely used. While the first one describes chemisorption, the Freundlich model describes physisorption [75].

Different kinetic models have also been studied to identify the critical mechanism by which the ion capture occurs to understand the mechanisms governing fluoride adsorption and isotherm models.

It has been reported that in the adsorption process, the mechanism is regulated by diffusion, and the adsorption rate is controlled by mass transfer [76].

The adsorption mechanism controls the fluoride adsorption capacity, energy, and kinetics [77].

Fluoride adsorption has been described as a three-step phenomenon:Molecular diffusion, external mass transfer, or fluoride ions transport from the aqueous medium to the external surface.Fluoride ion adsorption on the adsorbent particle surface.Intraparticle diffusion: the adsorbed fluoride ions are transferred to the inner surface of the adsorbent particle [78,79,80].

Figure 3 graphically illustrates the adsorption process described above.

Generally, the kinetic models described for fluoride adsorption can be classified into reaction-based and diffusion-based models. As with isotherm models, there is a wide variety of models involving different considerations [43,51,75,77,81,82,83]. Usually, the pseudo-first- and second-order models are applied in addition to the intra-particle diffusion model [77,84]. The pseudo-first-order model describes physical adsorption [76]. The pseudo-second-order model assumes a chemical adsorption process through the electron exchange between the fluoride and the adsorbent [82,85]. Finally, the intra-particle diffusion model assumes that the fluoride transport into the adsorbent particle pores is the rate-controlling step [86]. Table 2 summarizes the kinetic and isotherm models reported for fluoride bioadsorption by non-carbonized lignocellulosic-based bioadsorbents. The Langmuir isotherm and the pseudo-second-order model are most frequently reported for this bioadsorbent type. Therefore, as reaction-based models dominate, ion exchange or species replacement is reported as the leading mechanism for fluoride adsorption.

## 8. Advantages and Disadvantages of Biosorbents

Lacson et al. (2021) stated that adsorption, precipitation, and membrane-based processes are the most used methods to remove fluoride from water. The authors compared and evaluated these methods of fluoride removal, considering that all have their drawbacks. These include the operation complexity, high chemical consumption, high operating costs, and the large amount of by-products generated (e.g., sludge). Conceptual estimations of operating costs (OpEx) were then made. The results showed that flocculation coagulation (0.184 €/m^3^) was the lowest-cost technology, followed by electrocoagulation (0.230 €/m^3^). The most expensive treatment method was ion exchange resins, which cost 4675 €/m^3^. Adsorption using granular activated carbon (the most commonly used adsorbent) was estimated at 3740 €/m^3^, ranking among the most expensive technologies. The authors also concluded that adsorption is a suitable technique for defluoridation of drinking water sources around 20 mg fluoride/L or less [87].

Precipitation methods have not been thoroughly developed in recent years, and the disposal of the by-products generated is an issue of concern. Nevertheless, these techniques have demonstrated efficient removal capabilities in water with around hundreds of mg fluoride/L concentrations. Finally, membrane treatments achieve high fluoride removal efficiencies and have been widely developed commercially. However, they require high pressure and high energy consumption during operation, which is costly. Therefore, they are not economically attractive for water defluoridation in poor communities [87].

It is important to note that the use of waste lignocellulosic materials arises as a potential alternative for defluoridation due to the need to reduce costs in removing the contaminants without sacrificing the efficiency or quality of the treated water. In this regard, Manna et al., 2018 estimated costs in USD/kg of fluoride removed using different adsorbent materials. However, the adsorbent processing costs, energy, regeneration, and labor were not considered in the final price. The most commonly used materials for fluoride adsorption, such as activated metal oxides, activated carbon, and activated alumina, reached 8, 2, and 1.5 USD/kg of fluoride removed, respectively. Among the biomaterials, chitosan, algal, and fungal dry mass cost were estimated at 1800 and 0.5–6 USD/kg of fluoride removed, respectively. Lignocellulosic and chemically treated lignocellulosic plant matter was estimated at 0.3 and 0.5 USD/kg fluoride removed, respectively. The latter two are the lowest-cost adsorbent materials [32]. Few authors have developed economic feasibility studies. This may be attributed to biomaterial early stages of development and application [35].

The literature documents some differences between bioadsorbents and inorganic-based adsorbents. Table 5 shows the most relevant information found so far.

Biosorption has the advantages of adsorption and is an environmentally friendly option at a low cost. This advantage, however, may be clouded by the low performance of some studied materials and the need to use higher doses or undergo enhancement procedures. Even so, the use of lignocellulosic biomasses is a promising option for pollutant removal in aqueous media due to the possibility of reusing or giving value to waste materials, which would otherwise become waste to be disposed of.

## 9. Conclusions

The literature review presents numerous successful studies in developing bioadsorbents for fluoride reduction or removal from drinking water. The extensive research includes evidence of good adsorption capacities and the certainty of not leaving chemical traces (leaching of elements) that pose other risks. Furthermore, bioadsorbent regeneration and reuse studies are shown, and most importantly, the validation of the materials when used in groundwater or other real matrices.

The progress and improvement of analytical techniques have made it possible to infer the probable mechanisms and critical steps for fluoride sequestration by bioadsorbents. Understanding how and why such a process occurs allows for refining methodologies, preparation, and anion capture. It also predicts the bioadsorbents’ stability under different operating conditions.

Since non-carbonized lignocellulosic-based bioadsorbents are competitive in fluoride capture at different concentrations and conditions, it is essential to explore their application on larger scales and include economic feasibility studies.

## Figures and Tables

**Figure 1 polymers-14-05219-f001:**
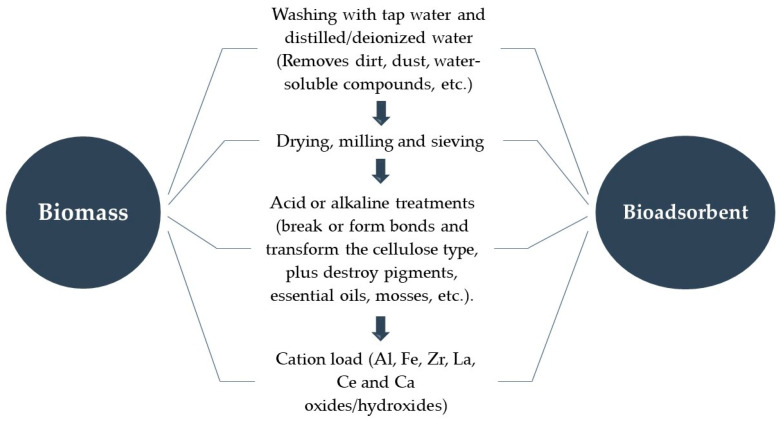
General processing of lignocellulosic biomass as bioadsorbents.

**Figure 2 polymers-14-05219-f002:**
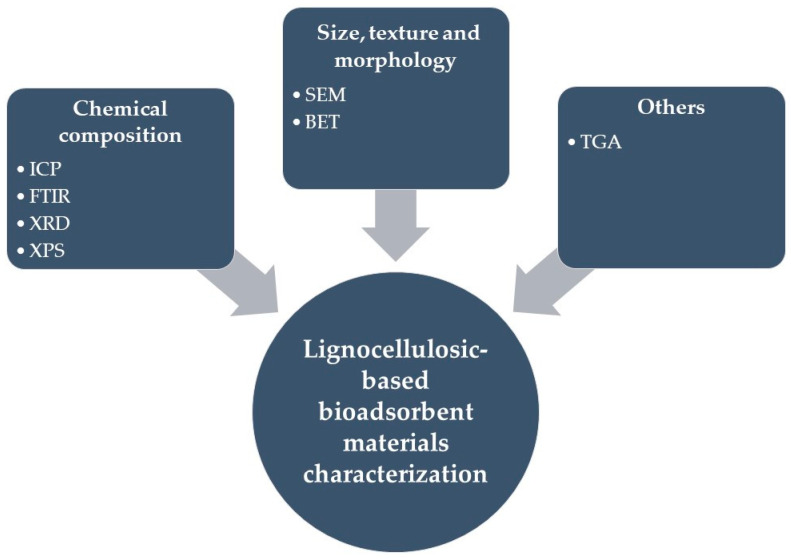
Bioadsorbent characterization from non-carbonized lignocellulosic biomasses.

**Figure 3 polymers-14-05219-f003:**
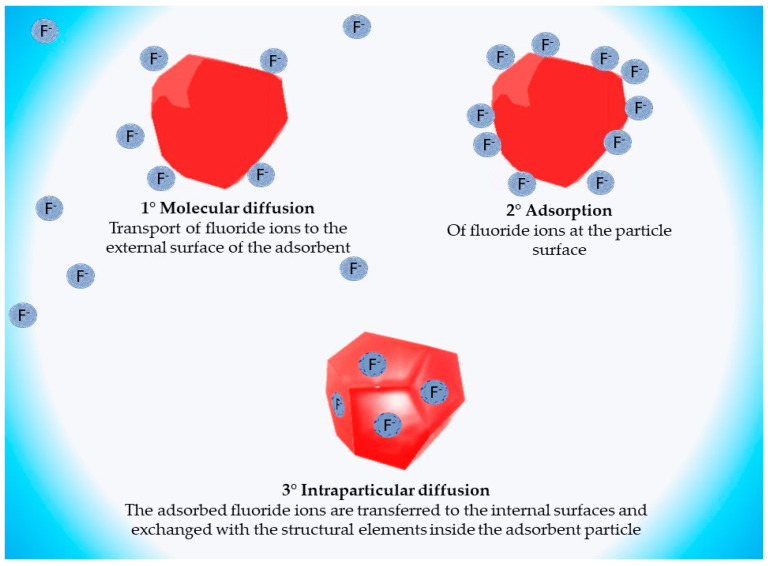
Fluoride adsorption process steps.

**Table 1 polymers-14-05219-t001:** Fluoride removal-reduction technologies in drinking water.

Technology	Description	Advantage	Disadvantage	Reference
Coagulation-precipitation				
Chemical precipitation Electrocoagulation	When chemicals were added, the suspended charged particles were then neutralized and agglomerated to settle down.	Good efficiency. Easy to use. Continuous or batch operation (for small flows). Simple design. Low cost.	Lack of ability to reduce fluoride below WHO limits. It requires the removal of bulky and wet sludge. Secondary treatment is needed. It is necessary that a high conductivity of the water be treated. Species dissolution and by-product formation.	[4,6,8,32,33]
Membrane-based processes				
Reverse osmosis Ultrafiltration Nanofiltration Electrofiltration Dialysis	Water is forced through a semi-permeable membrane to separate contaminants.	Production of high-purity water. High efficiencies. Automatic control. Selectivity.	Relatively expensive to install and operate. Susceptible to membrane fouling and degradation. Operates at high pressures. Significant energy demands. Requires water re-mineralization and pH adjustments.	[4,11,33,34]
Ion Exchange and adsorption				
Ion Exchange resins Chelating resins Adsorbents: activated carbons Metal oxides Nanomaterials Biomaterials Industrial wastes	Process in which the ions in aqueous media are transferred to the adsorbent matrix by several mechanisms. It includes physical adsorption or chemisorption through chelation, complexation, and ion exchange processes.	It allows the adsorbent material regeneration. High removal capacity. Anion selective removal. Low cost.	Highly pH-dependent. Vulnerable to interference.	[4,6,8,9,31,32]

**Table 2 polymers-14-05219-t002:** Studied biomasses for fluoride removal in drinking water.

Biomass	Removal (%)	pH	Removal Capacity (mg/g)	Contact Time (min)	Adsorbent Dosage (g/L)	Isotherm	Kinetic Model	Mechanism	Anion Interference
Zr-loaded orange residues [38]	--	2.4	22.8	480	1.6	L	--	Ligand exchange	NO_3_^−^ > SO_4_^−2^ > Cl^−^ > CO_3_^−2^
Orange waste loaded with La, Sm, Sc, Ho, and Lu [49]	--	4.0–6.0	14.6–21.8	1140	1.6	L	--	Ligand exchange	--
Sawdust and wheat straw [39]	49.8 and 40.2	6.0	1.7 and 1.9	60	4.0	F	PSO, IPDM	Bond formation, surface adsorption, and intraparticle diffusion.	--
Lemon leaves [40]	70	2.0	-	145	200	--			--
Tea wastes loaded with Al-Fe oxides [43]	90	4.0–8.0	3.8–18.5	120	2.0	L	PSO	Covalent Bond Formation/Ligand exchange	--
Elephant grass and water hyacinth [50]	85	4.0	7.0 and 5.0	210	1.5 and 1.0	L,F	IPDM	--	--
Jute fibers [44]	98	5.0	5	120	1.5	L	PSO	OH^−^ replacement by F^−^	SO_4_^−2^ > HCO_3_^−^ > CO_3_^−2^ > NO_3_^−^ > PO_4_^−3^
Zr- orange peels [51]	97	7.0	5.6	50	0.7	L	PSO	Ligand exchange	PO_4_^−3^ > HCO_3_^−^> NO_3_^−^ > SO_4_^−2^ > Cl^−^
Pineapple peels [52]	90	4.0	-	60	0.6	F	E	Electrostatic attraction	--
Al-modified pine sawdust [14]	59.5	6.0	3.6	120	0.5	L	E	OH^−^ and F^−^ ions exchange	--
Colocasia esculenta stem [53]	33	4.25	-	180	20	--	--	--	--
Banana peels [41]	90	4.0	1.2	60	1.5	D-R	B	--	CO_3_^−2^ > PO_4_^−3^ > SO_4_^−2^ > NO_3_^−^ > Cl^−^
La-modified cellulose extracted from sugar cane bagasse [54]	98	3.0	1.1	60	2.5	L,F	PSO	Cl exchange with F^−^	Cl^−^ > NO_3_^−^ > SO_4_^−2^ > HCO_3_^−^
Tea waste with Al [55]	52.9	5.2	3.2	60	2.0	F	--	Ion exchange	--
Melon peels [56]	90	7.0	3.0	50	5	L	PSO	--	SO_4_^−2^ > Cl^−^ > Br^−^ > NO_3_^−^
Ziziphus leaves [57]	95.3	7.0	0.48	25	5	L	PFO	--	--
Al-modified guava seeds [45]	80	6.0	0.3	150	70	L,F	PSO	--	--
Coffee beans [58]	89	4.0	9.0	105	2	L	PSO	--	--
Fe-impregnated sisal fiber [46]	53.4	2.0	0.2	60	15	L	--	Electrostatic interactions, and exchange of ligands	--
Zr-modified grape bagasse [42]	90	3.0	7.54	60	6	L	--	Ligand exchange	HPO_4_^−2^ > CO_3_^− 2^ > NO_3_^−^ > Cl^−^ > SO_4_^−2^
Prosopis cineraria and Syzygium cumini leaves [59]	--		11.5 and 7.4	120 y 90	1	L	PSO	Film diffusion	--
Sugar cane bagasse and fruit husks [60]	84 and 78	6.0 and 4.0	--	100	12 and 10	--	--	--	--
Tamarind seed husk [48]	94	6.0	1.79	60	0.3	L	PSO	--	--
Flower petals [61]	80	6.0	1.29	90	2.5	L	PSO	--	--
CeO_2_ -modified wood waste [37]	--	3.0	48	120	--	F	PSO	OH^−^ replacement by F^−^	--
Zr-loaded garlic husks [47]	92	--	--	--	--	--	--	--	--

PFO: Pseudo-first-order model, PSO: Pseudo-second-order model, IPDM: Intraparticle diffusion, B: Bahangam, E: Elovich, L: Langmuir, F: Freundlich isotherm and D-R: Dubinin–Radushkevich.

**Table 3 polymers-14-05219-t003:** Lignocellulosic bioadsorbents: physical and chemical treatments.

Biomass	Drying Temperature (°C)	Grind	Particle Size (μm)	Acidification	Alkalinization	Middle Treatment	Cation Charge
Zr-loaded orange residues [38]	Oven, 70	Mortar	100–150	--	Ca (OH)_2_ @ 30 °C, 24 h, pH 12 with NaOH	--	0.1 M de ZrOCl_2_·8H_2_O at 30 °C, 24 h, pH 2.2
Orange waste loaded with La, Sm, Sc, Ho, and Lu [49]	Oven, 70	--	100–150	--	Ca (OH)_2_ @ 30 °C, 24 h, pH 12 with NaOH	--	0.1 M of Sc^+3^, La^+3^, Sm^+3^, Ho^+3^, and Lu^+3^, at 30 °C, 24 h, pH 2.2
Sawdust and wheat straw [39]	Solar/oven, 80	--	300–850	Formaldehyde treatment l 1% at 50 °C, 24 h			
Lemon leaves [40]	Solar	Mortar	1500	HNO_3_ 1 M, heating @ 20 min	NaOH 0.5 M, heating @ 20 min	--	--
Tea wastes are loaded with Al-Fe oxides [43]	Oven, 70	--	250	H_2_SO_4_ 0.02 M @ 70 °C, 5 h	--	--	0.1 M of FeCl_3_, 0.4 M Al (NO_3_)_3_ at 60 °C, pH 5.0 with NaOH (2 M) @ 30 min
Elephant grass and water hyacinth [50]	70	Mechanical	150	--	0.5% p/v NaOH @ 30 °C, 24 h. Steamed at 103 kPa	--	--
Jute fibers [44]	85	Grinder	300		0.5% *w*/*v* NaOH @ 30 °C, 30 min. Steamed at 103 kPa, 121 °C @ 30 min.	Alkaline aqueous emulsion of neem oil and phenolic resins at 105 °C @ 1 h.	
Zr-orange peels [51]	Oven, 50	--	100–150	--	NaOH 0.1 M @ 24 h	0.1 M ClCH_2_CO_2_H, pH 8–10 with NaOH 0.1 M @ 24 h	0.1 M ZrOCl_2_·8H_2_O, 48 h
Colocasia esculenta stem [53]	Oven, 110		250	--	--	--	--
Al-modified pine sawdust [14]	Oven, 50	--	500	--	--	--	AlCl_3_ 0.05 M at pH 3.5 @ 3 h.
Banana peels [41]	Oven, 50	Grinder	200	--	--	--	--
La-modified cellulose extracted from sugar cane bagasse [54]	Solar	--	--	CH_3_COOH/HNO_3_	--	--	Dispersion in methanol and sonication with LaCl_3_ for 20 min.
Tea waste with Al [55]	Aire, 70	--	300	--	--	--	AlCl_3_ 0.3 M, HCl 0.01 M, NaOH 2 M at 60 °C, pH 5.5
Melon peles [56]	Oven, 70	--	75	--	--	--	--
Ziziphus leaves [57]	Oven, 105	--	710	--	--	--	--
Al-modified guava seeds [45]	Oven, 60	--	1000	0.5 M HCl at 70 °C @ 20 min.	--	--	AlCl_3_ 0.05 M @ 3 h.
Coffee beans [58]	Oven, 110	--	75	HCL hot water bath @ 2 h.	--	--	NaOH at pH 12 @ 1 min.
Zr-modified grape bagasse [42]	Oven, 60	Grinder	425	--	--	--	0.1 M ZrOCl_2_·8H_2_O, pH 1.35 @ 24 h
Zr-loaded garlic husks [47]	Oven, 60	--	425	--	--	--	0.1 M ZrOCl_2_·8H_2_O, pH 1.2 @ 24 h
Prosopis cineraria and Syzygium cumini leaves [59]	Oven, 105	Mill	710	--	--	--	--
Sugar cane bagasse and fruit husks [60]	Solar	Grinder	--	--	--	--	--
Tamarind seed husk [48]	Oven, 150	Mortar	100	--	--	--	--
Flower petals [61]	Solar/oven, 70	--	--	--	--	--	--
CeO_2_ –modified wood waste [37]	Oven, 80	--	--	--	NaOH 5% at 80 °C @ 4 h, NaClO_2_ 5% at pH 4.0 and 80 °C @ 5 h.	NaOH + CH_4_N_2_O at −12.8 °C, ultrasound with PVA @ 4 h.	--

**Table 4 polymers-14-05219-t004:** Groundwater’s main ions.

Main Ions	
Cation	Anion
K^+^	HCO_3_^−1^
Na^+^	Cl^−1^
Ca^+2^	NO_3_^−1^
Mg^+2^	CO_3_^−2^
	SO_4_^−2^

**Table 5 polymers-14-05219-t005:** Comparison of several adsorbents.

Parameter	Bioadsorbents	Inorganic-Based Adsorbents
Cost	Low-cost, discarded materials are used. Household waste, agricultural waste, and even some agro-industrial waste are included.	Usually, they are more expensive.
pH	Operating ranges have been reported from 2 to 7, with no difference with inorganic adsorbents.	Broad range of operation, generally working from 2–7 as well as bioadsorbents.
Reuse	Up to 10 desorption cycles have been reported; some suggest direct material disposal.	They are more frequently reported, and certain materials have the potential to regenerate more than 5 cycles, with no significant decrease in their removal capacity.
Removal capacity	Low capacity often has to be modified.	High capacities, working alone or in conjunction with other cations.
[9,11,20,24,31,35,36,78]		

## Data Availability

Data are available only upon request to the authors.

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
