# Peer review of "Lignocellulosic Biomass as Sorbent for Fluoride Removal in Drinking Water"

_polymers, 2022, doi:10.3390/polym14235219_

Round 1

Reviewer 1 Report

Dear Authors

The submitted review for publication is very crucial since it treated an essential issue for the human life quality.

The water supply to millions of people worldwide is of alarmingly poor quality. Supply sources are depleting, whereas demand is increasing. Health problems associated with water consumption exceeding 1.5 mg/L of fluoride are a severe concern for the World Health Organisation (WHO). Therefore, it is urgent to research and develop new technologies and innovative materials to achieve partial fluoride reduction in water intended for human consumption. The new alternative technologies must be environmentally friendly and be able to remove fluoride at the lowest possible costs. So, the use of waste from lignocellulosic biomasses provides a promising alternative to commercially inorganic-based adsorbents—published studies present bioadsorbent materials competing with conventional inorganic-based adsorbents satisfactorily. However, it is still necessary to improve the modification methods to enhance the adsorption capacity and selectivity, as well as the reuse cycles of these bioadsorbents.

The topic is interesting and the work in general is well presented.

However, I have some comments here;

1- The section on the "Biosorbents characterization" is not relevant to the review topic.

2- A cost-effective study should be included to show the economic benefits of using the lignocellulose adsorbents. 

3- A comparison with the commercially available system of Flouride anions removal is essential.

A revision is needed before considering the publication of the submitted review. 

Reviewer 2 Report

Review of paper ‘Lignocellulosic biomass as sorbent for fluoride removal in drinking water’ prepared by Adriana Robledo-Peralta, Luis A. Torres-Castañón, René I. Rodriguez-Beltrán, and Liliana Reynoso-Cuevas.

The manuscript polymers-1969353 is a brief review on the use of biomass in water treatment. I have some suggestions that authors may consider before publishing this work:

1. The authors describe 22 different biomasses in tables, but the same in all tables, duplicating some of the information, including their names and references. In my opinion, these tables should be combined as much as possible to avoid duplication of information.

2. The paper lacks a conclusion on the aspects discussed. For example, the authors describe forms of chemical treatments on lignocellulosic bioadsorbents (see Table 4), but do not indicate the desired treatment directions. The authors should, on the basis of the studies cited, indicate the advantages and disadvantages of the methods proposed in the literature, possibly indicate other possibilities, etc. All work requires in-depth discussion.

3. It is completely inaccurate to describe the basic information on research methods. For example, describing the ICP technique, including the method of measurement and the possibility of determining many elements, is, from the point of view of the subject of the publication, superfluous. It should be assumed that the readers are familiar with the basics of the techniques described. The following passages should be deleted:

-lines 209-224 (ICP)

- lines 239-249 with Figure 4 (TGA)

- lines 258-264 (SEM)

- lines 295-311 (BET)

- lines 326-346 (FTIR)

- lines 380-391 (XRD)

- lines 409-421 (XPS)

At the same time, the authors should, in addition to indicating that determinations have been made in the cited publications, refer to how these characteristics should contribute to the development of sorbents in the context of written water treatment.

Furthermore, this description of methods mostly quotes Jha's work: after all, one can find other works in this field. Basing yourself on one is not a review

3. Editing errors:

- in Table 2, the ‘y’ should be changed to ‘–‘,

- Figure 1: the drawing is blurred and the font is too small,

- Figure 1: the drawing is blurred.

Reviewer 3 Report

The review article "Lignocellulosic biomass as sorbent for fluoride removal in drinking water" is presented on a hot topic: purification of drinking water using pre-treated lignocellulose from fluorine-containing compounds. This material is based on 97 references, but is made very high quality. The article briefly but exhaustively describes the technologies for removing fluorine from water, biosorption as a process, various sources of lignocellulose, methods for transforming these sources into biosorbents, gives a more detailed assessment of the physical and chemical methods for transforming lignocellulose into biosorbents, and a whole chapter is devoted to research methods. It is very important that the authors also described the mechanism of fluorine sorption and provided an exhaustive table 8 with models for non-carbonized lignocellulose. There is no doubt that this review is necessary for readers, since it provides the author's interpretation of the ideas on this topic, without excluding further development and discussion.

I think that the review could be published in the form presented.

I have two small remarks that I will make for the authors:

1. Pay attention to Figure 1: In the caption it is indicated that “General processing of lignocellulosic biomass as bioadsorbents”, and in the figure itself, the transformation of biomass into a “biocomposite”. This is a missprint?

2. Check the design of links, there are inconsistencies, for example, "Polymers" does not require doi.

Round 2

Reviewer 1 Report

Dear Authors

Thank you very much for considering the raised comments during your revision process.

I can recommend the revised version of your review for publication. 

Reviewer 2 Report

I have no further comments.